# Fidelity of hyperbolic space for Bayesian phylogenetic inference

**Matthew Macaulay** [1]ᵒ*, **Aaron Darling**[2]ᵒ, **Mathieu Fourment**[1]ᵒ

**1** University of Technology Sydney, Australian Institute for Microbiology & Infection, Sydney, Australia,
**2** Illumina Australia Pty Ltd, Sydney, Australia

ᵒ These authors contributed equally to this work.
\* matthew.macaulay@uts.edu.au

## Abstract

Bayesian inference for phylogenetics is a gold standard for computing distributions of phylogenies. However, Bayesian phylogenetics faces the challenging computational problem of moving throughout the high-dimensional space of trees. Fortunately, hyperbolic space offers a low dimensional representation of tree-like data. In this paper, we embed genomic sequences as points in hyperbolic space and perform hyperbolic Markov Chain Monte Carlo for Bayesian inference in this space. The posterior probability of an embedding is computed by decoding a neighbour-joining tree from the embedding locations of the sequences. We empirically demonstrate the fidelity of this method on eight data sets. We systematically investigated the effect of embedding dimension and hyperbolic curvature on the performance in these data sets. The sampled posterior distribution recovers the splits and branch lengths to a high degree over a range of curvatures and dimensions. We systematically investigated the effects of the embedding space's curvature and dimension on the Markov Chain's performance, demonstrating the suitability of hyperbolic space for phylogenetic inference.

**Data Availability Statement:** The software presented in this article is open source and freely available from https://github.com/mattapow/dodonaphy. The initial release is also stored on Zenodo doi: 10.5281/zenodo.6667956, available at

## Author summary

### Why was this study done?

- Tree structures are widely used in fields such as phylogenetics, however modifying the layout and branch lengths of these structures simultaniously is a high-dimensional problem.

- Recent work in machine learning has demonstrated the usefulness of representing tree-like data as points in low dimensional hyperbolic space.

- We aimed to explore new ways of representing phylogenetic trees so they can be modified in a continuous manner.

https://doi.org/10.5281/zenodo.6667956. All analysed data sets have been previously published and freely available from Dryad https://doi.org/10.5061/dryad.6hdr7sr3p (DOI: 10.5061/dryad.6hdr7sr3p).

**Funding:** This work was supported by the Australian Government through the Australian Research Council awarded to MF. Project number LP180100593. https://www.arc.gov.au/ The funders had no role in study design, data collection and analysis, decision to publish, or preparation of the manuscript.

**Competing interests:** The authors have declared that no competing interests exist.

## What did the researchers do and find?

- We represented trees by the locations of their embedded genomic sequences in hyperbolic space.

- We perturbed these continuous encoding locations and decoded an altered discrete tree structure.

- Using this technique, we performed Bayesian inference and computed the posterior distribution of standard eight datasets, to demonstrate the feasibility of phylogenetic inference with this representation.

- We found that hyperbolic space is suitable for Bayasian phylogenetics and is most efficient across a broad range of hyperbolic curvatures with low dimensionality.

## What do these findings mean?

- This method diversifies the way numerical methods can navigate the space of trees both in phylogenetics and more broadly.

- With hyperbolic embeddings, scaleable online inference is possible by quickly adding taxa to a tree or a distribution of trees.

- This method could open a wealth of powerful continuum-based methods to navigate the space of trees.

This is a *PLOS Computational Biology* Methods paper.

## Introduction

Bayesian phylogenetics seeks the posterior distribution over discrete tree topologies and continuous tree branch lengths and evolutionary model parameters given an alignment of nucleotide sequences. Computing the posterior is analytically intractable so it is approximated using Markov chain Monte Carlo (MCMC) [1, 2]. MCMC is a workhorse algorithm of Bayesian phylogenetics that works by drawing autocorrelated chains of samples from the posterior distribution. New samples are proposed "nearby" to the current state of the Markov chain. However, the space of phylogenetic trees is super-exponential in the number of topologies and navigating tree space is difficult [3, 4]. Most methods to propose a new tree topology are relatively simplistic, for example nearest neighbour interchange (NNI) changes the connection between just four nodes. Subtree prune and re-graft (SPR), which removes a clade and re-attaches it to a new position, can make more substantial changes. However, SPR and NNI exporation may face difficulties scaling up to make substantive changes to a topology in the high dimensional space of trees. Improvements on these by guided proposals comes at the cost of additional computation [5]. This all poses the dilemma of how to best represent and navigate the space of phylogenies.

Recent work in machine learning has demonstrated how hyperbolic space offers quality embeddings of tree-like data in low dimensions. Notably, hyperbolic representations have successfully clustered data hierarchically into trees [6–8]. These works optimise a carefully devised objective function to find an optimal embedding. However, unlike, these methods, the objective function for Bayesian phylogenetics is prescribed by the model of evolution and prior distribution. Nonetheless, embedding genomic sequences as points in hyperbolic space and working in the embedding space could provide a way to move through tree space with an improved notion of "locality"—small changes to the embedding locations of sequences produce small changes in their distances on the tree. The embedding may enable a more natural representation of sequence divergences, potentially enabling an MCMC sample to make more complex changes to the model (topology, branch lengths etc) in a single move. Recent works reviewed by [9] have used representations, or specifically hyperbolic representations to learn a maximum likelihood phylogeny [10, 11]. However, their Bayesian counterpart is missing.

Bayesian phylogenetic practitioners increasingly need scalable methods to deal with larger sets of sequences [12], as highlighted by the current global pandemic of SARS-CoV-2 [13]. A demonstration that MCMC works in hyperbolic space could open up a wealth of possibilities for working with embeddings for Bayesian phylogenetics. As opposed to previous use of machine learning techniques in phylogenetics, our work decodes a tree and uses the phylogenetic posterior probability as a cost function for hyperbolic embeddings. It also generates a distribution of phylogenetic trees rather than a point estimate.

To assess whether hyperbolic embeddings can represent a posterior distribution of trees, we perform MCMC on trees decoded from an embedding of nucleotide sequences. Trees are decoded using the neighbour-joining (NJ) algorithm [14] before computing their likelihood and prior probability for each MCMC generation. We implement this MCMC in a python package called Dodonaphy, which is available on GitHub [15].

The goal of this paper is to empirically demonstrate the fidelity of hyperbolic embeddings for Bayesian phylogenetics. We begin by detailing the concepts from Bayesian phylogenetics, hyperbolic embeddings of trees and MCMC that are necessary to present the method. Once devised, we investigate the fidelity of the method and illustrate the embedding landscape. We then quantify the effects of the curvature and dimension of the embedding space on the MCMC performance. We also explore how a proposal distribution in the embedding space transfers to tree space. Finally, we discuss the numerous future research possibilities that this method opens.

## Materials and methods

This methods section introduces Bayesian phylogenetics and hyperbolic space before explaining how they can be combined in an algorithm. It discusses the embedding of an initial phylogenetic tree in hyperbolic space and decoded using the neighbour-joining algorithm. The uniqueness and continuity properties of the decoding process are assessed for performing MCMC on a number of datasets.

### Bayesian phylogenetics

In phylogenetics, the posterior distribution is the joint posterior distribution of the tree and evolutionary model parameters given an alignment of nucleotide sequences $Y$. We refer to a phylogenetic tree $T$ which includes both the topology and branch lengths of the tree. The

posterior probability of a tree is

$$p(T|Y) = \frac{p(Y|T)p(T)}{p(Y)}.$$

Dodonaphy uses a simple model of evolution, the Jukes-Cantor (JC69) model [16] to compute the likelihood given a tree $p(Y|T)$ using the Felsenstein pruning algorithm [17]. The JC69 model is a continuous Markov chain where each DNA character has an equal probability of evolving into another character.

For the prior $p(T)$, we utilise a uniform distribution over tree topologies and a Gamma-Dirichlet prior probability of a tree's branch lengths, as previously suggested by [18]. The Gamma-Dirichlet prior assigns a Gamma distribution $\Gamma(\alpha, \beta)$ with shape $\alpha = 1$ and rate $\beta = 0.1$ on the total tree length (sum of branch lengths), before dividing this into individual branches with a Dirichlet distribution $\text{Dir}(\alpha_k = 1.0)$.

The state-of-art phylogenetic software MrBayes [19] offers both of these models for the prior and likelihood, which allows us to directly compare its results with Dodonaphy. We use two *golden* runs of MrBayes with $10^9$ iterations and consider these the *ground truth* posterior distribution.

We approximate the posterior distribution through MCMC sampling, which does not require computing the intractable probability of the data $p(Y)$. We use Metropolis-Coupled MCMC with four chains and an adaptive multivariate normal proposal. A key element of MCMC for phylogenetics is proposing a "nearby" tree state, for which, we move to a hyperbolic embedding.

## The Hyperboloid model

One common model of Hyperbolic space in $d$ dimensions is the upper sheet of a hyperboloid $\mathbb{H}^d = \{\boldsymbol{x} \in \mathbb{R}^{d+1} : \langle \boldsymbol{x}, \boldsymbol{x} \rangle = -1\}$ where the Lorentz inner product is defined as

$$\langle \boldsymbol{x}, \boldsymbol{y} \rangle = \boldsymbol{x}^\mathsf{T} H \boldsymbol{y}, \qquad H = \begin{bmatrix} -1 & 0 \\ 0 & I \end{bmatrix} \in \mathbb{R}^{d+1} \times \mathbb{R}^{d+1}.$$

For example, with one dimension $d = 1$ the hyperbola of points $(x, y) \in \mathbb{R}^2$ that satisfy $(x, y)H(x, y)^\mathsf{T} = -x^2 + y^2 = 1$ simplifies to $y^2 = 1 + x^2$. Note that whilst the sheet sits in $\mathbb{R}^{d+1}$, it is still a $d$-dimensional object, just as a 1D hyperbola is a line sitting in $\mathbb{R}^2$. The first coordinate is determined by the last $d$ coordinates according to $\langle \boldsymbol{x}, \boldsymbol{x} \rangle = -1$ to stay on the sheet. This rearranges to give $x_0 = \sqrt{x_1^2 + x_2^2 + \ldots x_d^2 + 1}$. Thus we need only store the last $d$ points in $\mathbb{R}^d$ and project up exactly onto the hyperboloid by $\phi : \mathbb{R}^d \to \mathbb{H}^d \subset \mathbb{R}^{d+1}$. Explicitly, we use:

$$\phi(\boldsymbol{x}) = \begin{bmatrix} x_0 \\ \boldsymbol{x} \end{bmatrix} \tag{1}$$

projecting the first coordinate and leaving the rest unchanged, Fig 1. The inverse mapping omits the first coordinate: $\phi^{-1}([x_0, x_1, \ldots, x_d]^\mathsf{T}) = [x_1, \ldots, x_d]^\mathsf{T}$.

The length of the geodesic between two points on the sheet is given by the metric

$$d_\kappa(\boldsymbol{x}, \boldsymbol{y}) = \frac{1}{\sqrt{-\kappa}} \text{arcosh}(-\langle \boldsymbol{x}, \boldsymbol{y} \rangle).$$

In this model, for a fixed distance $d_\kappa(\boldsymbol{x}, \boldsymbol{y})$, a more negative curvature $\kappa < 0$ stretches how far apart the points are on the curved sheet, imparting more curvature between the points.

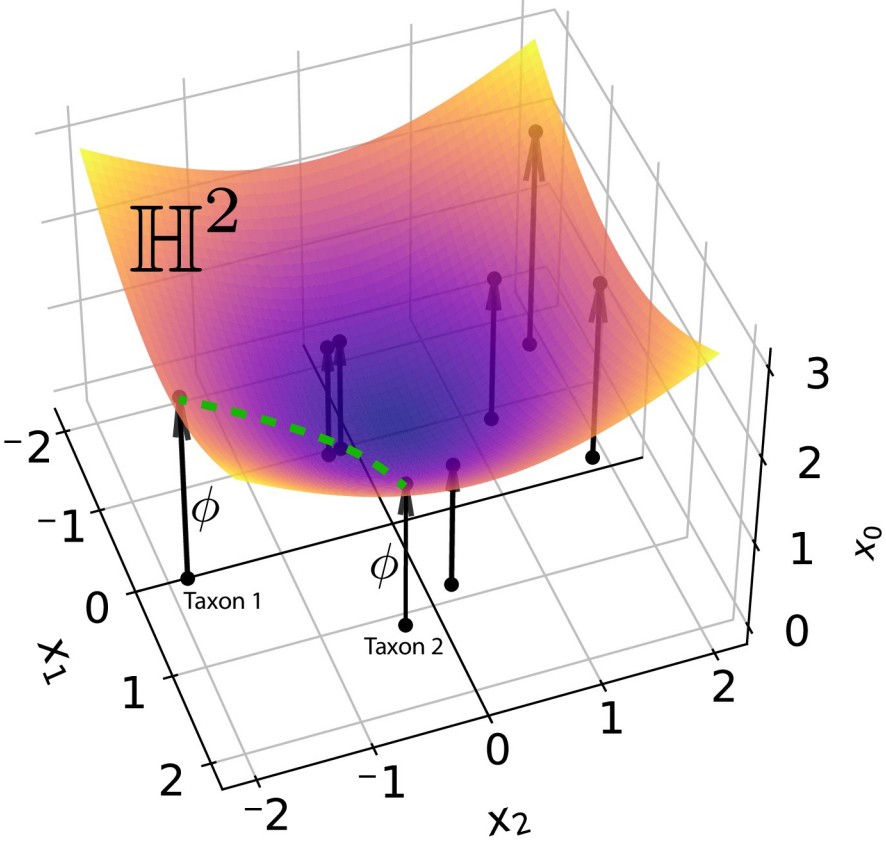

**Fig 1. Hyperbolic embedding of eight taxa.** Eight points (taxa) lie on a two-dimensional hyperboloid sheet $\mathbb{H}^2$ in $\mathbb{R}^3$. Arrows indicate how each two-dimensional point $(x_1, x_2)$ is deterministically projected up onto the sheet by $\phi$. The green dashed line illustrates that the distance between two taxa is the length of the geodesic on the hyperboloid surface.

Conversely, to maintain a fixed distance as $\kappa \to 0$, points move towards the origin where the space is flatter. The effects of curvature $\kappa$ and dimension $d$ will be investigated empirically on a number of real datasets.

## Hyperbolic embeddings (Encoding)

To initialise an embedding of the taxa, Dodonaphy begins by minimising the stress of the embedding to a distance matrix. The distance matrix may either derive from a provided starting tree or the genetic distances between the aligned sequences. The stress of an embedding of points $\boldsymbol{x}_i \in \mathbb{H}^d$ to a given set of pairwise distances $D_{ij}$ is

$$\sigma^2 = \sum_{i,j} (D_{ij} - d(\boldsymbol{x}_i, \boldsymbol{x}_j))^2.$$

Minimising this stress is a challenging non-convex problem, however, Hydra+ is a recent algorithm that provides a fast approximate solution [20]. It begins by finding a set of solution locations to the much simpler corresponding strain minimisation problem through an Eigendecomposition. Then it minimises the stress through gradient-based optimisation. We reimplemented Hydra+ as a Python package available on GitHub at https://github.com/mattapow/hydraPlus.

We embed the $n$ labelled taxa as a set of points in a hyperbolic space $X = \{x_i\}$, $i = 1, 2, \ldots, n$ using hydra+. By representing taxa as points on the hyperboloid $x_i \in \mathbb{H}^d$ we work directly with their hyperbolic distances $d_\kappa(x_i, x_j)$.

## Decoding trees from embeddings

Dodonaphy then employs neighbour-joining to decode a tree $T = \text{NJ}(D)$ from the pairwise hyperbolic distance $D_{ij} = d(\phi(x_i), \phi(x_j))$ of a set of embedding locations $X$. This decodes both branch lengths and a tree topology at once. The NJ algorithm takes pairwise distances as input, which is efficiently computed in the hyperboloid model of hyperbolic space using linear algebra [21]. NJ works by recursively agglomerating the closest two taxa based on their pairwise distances (and is adjusted for their proximity to all other taxa).

The main advantage of using NJ is its consistency: it correctly constructs a tree when the given distances fit on that tree. However, instead of the sequence distances, Dodonaphy utilises the hyperbolic distances between taxa for NJ. This allows Dodonaphy to move through tree space by moving the set of $n$ tip vectors $X$ through hyperbolic space $\mathbb{H}^{n \times d}$.

**Unique tree decoding.** Working in a general embedding space can produce arbitrary distance matrices that, problematically, may not be consistent with a tree structure. However, as we will see, the distances in hyperbolic space can be coerced to be arbitrarily close to a neighbour-joining tree. The neighbour-joining algorithm fits a unique unrooted tree provided that the pairwise distances satisfy the four-point condition: $\forall w, x, y, z$

$$d(x, w) + d(y, z) \leq \max\{d(x, y) + d(z, w), d(x, z) + d(y, w)\}.$$

When this occurs for all pairs of points, the distances are said to be additive and a tree consistent with the given distances can be recovered by NJ.

Hyperbolic space is often chosen for embedding trees because there exists a bound $\delta > 0$ on how much hyperbolic distances violate the four-point condition:

$$d(x, w) + d(y, z) \leq \max\{d(x, y) + d(z, w), d(x, z) + d(y, w)\} + 2\delta. \quad (2)$$

Such spaces are called $\delta$-hyperbolic. Furthermore, this bound $\delta$ can be scaled to become arbitrarily small by making the curvature $\kappa$ more negative [10].

**Tree length continuity.** The $\delta$-hyperbolic property of hyperbolic space makes it suitable for distance-based phylogenetics. We build on work by [10] to highlight the continuity of the decoded tree lengths. First, let $\delta_{d,\kappa}$ denote the minimal such error $\delta$ in the four point condition in $d$ dimensions with curvature $\kappa$, Eq 2.

**Theorem 1**. [10] *For any $d \geq 2$ and $\kappa < 0$, there exists $\delta > 0$ such that $\mathbb{H}^d$ with curvature $\kappa$ is $\delta$-hyperbolic. Furthermore,*

$$\delta_{d,\kappa} = \frac{1}{\sqrt{-\kappa}} \delta_{d,1}.$$

This scaling is significant for neighbour-joining because it leads to additive embedding distances. A distance matrix $D$ is *additive* if there is a weighted tree such that the distances on the tree $D_T$ match the given distances $D = D_T$. The distances are additive when the four-point condition is satisfied $\delta = 0$. When the input distances are additive, neighbour-joining decodes a consistent tree. That is, it decodes a unique tree with correct tree distances: $D = D_T$.

Neighbour-joining remains consistent when the distances are almost additive. For the following theorem, we need the $l_\infty$ norm, written $||D - D_T||_\infty$, which is the maximum element-

wise difference between $D$ and $D_T$. The radius of a method $f$ is the maximum $\alpha$ in

$$||D - D_T||_\infty < \alpha \min_{e \in E(T)}(l_e)$$

so that $f(D) = T$ produces the same tree with edges $E(T)$.

**Theorem 2.** [22] *The $l_\infty$ radius of neighbour joining is $\frac{1}{2}$.*

**Corollary 3.** *In the limit $\kappa \to -\infty$, the length of a decoded tree $l(NJ(D(X))$ is continuous in the embedding locations X.*

*Proof.* In the limit $\kappa \to -\infty$, the error becomes arbitrarily small $\delta_{d,\kappa} \to 0$. In particular, there exists $\kappa < 0$ such that $||D - D_T||_\infty < \alpha \min_{e \in E(T)}(l_e)$. When this occurs, theorem 2 ensures the correct topology is decoded and the distances are arbitrarily close to *additive*. Consequently, the hyperbolic distances match the tip-tip distances on the tree, even as the topology changes. Since the hyperbolic distances are continuous in the embedding locations, so are the tree lengths.

If $\kappa$ is too large, NJ may not reconstruct a unique tree, making the MCMC dependent on the particular taxa selected to merge in the algorithm.

## Dodonaphy algorithm

In summary, Dodonaphy embeds pairwise tip distances $D$ with hydra+ and performs phylogenetic MCMC on a hyperboloid, algorithm 1. The MCMC is run with a Gamma-Dirichlet prior on the branch lengths and a tree likelihood under a JC69 model of evolution. The MCMC runs four parallel chains and performs ten MCMC chain swap moves every $10^3$ generations.

**Algorithm 1** Hyperbolic MCMC algorithm in Dodonaphy.

```
 1: procedure DODONAPHY(Y, κ, d, iterations)
 2:    D ← Hamming distances(Y)              ▷ Pair-wise alignment distances
 3:    X_hyp ∈ ℍ^{n×d} ← hydra + (D, κ, d)           ▷ Embedding locations
 4:
 5:    X ∈ ℝ^{n×d} ← φ⁻¹(X_hyp)        ▷ Store in tangent plane
 6:    D_ij ← d_κ(φ(x_i), φ(x_j))            ▷ Pair-wise hyperbolic distances
 7:    T ← NJ(D)              ▷ Decode NJ tree
 8:    for iterations do
 9:       X* ← draw from 𝒩(X, Σ)        ▷ Propose new state
10:       D*_ij ← d_κ(φ(x*_i), φ(x*_j))           ▷ Pair-wise hyperbolic distances
11:       T* ← NJ(D*)              ▷ Decode NJ tree
12:       X, T ← X*, T* with probability α              ▷ Metropolis step
13:    end for
14: end procedure
```

To propose a new tree, Dodonaphy samples points in the space tangent to the hyperboloid at $\mathbf{0} = [1, 0, 0, \ldots]^\top$, we call this point the *origin* of the space. We denote the tangent to the origin as $\mathbb{T}_\mathbf{0}\mathbb{H}^d \equiv \mathbb{R}^d$. For each of the $n$ sequences, Dodonaphy samples one point from $\mathbb{T}_\mathbf{0}\mathbb{H}^d$. In general, for a point $\mu$ on the hyperboloid, the tangent space is the set of vectors on the plane in $\mathbb{R}^{d+1}$ that is tangent to the hyperboloid at $\mu$. These are the points orthogonal to $\mu$ in $\mathbb{R}^{d+1}$:

$$\mathbb{T}_\mu\mathbb{H}^d := \{x \in \mathbb{R}^{d+1} : \langle x, \mu \rangle = 0\}. \tag{3}$$

Since the tangent plane to the origin $\mathbb{T}_\mathbf{0}\mathbb{H}^d$ is isomorphic to Euclidean plane $\{\mathbf{x} \in \mathbb{R}^{d+1} | x_0 = 1\}$, Dodonaphy can sample from any distribution in Euclidean space.

These points are projected onto the hyperboloid by $\phi$, and then using the pairwise distances on the hyperboloid a NJ tree is formed. This results in a tree $T$ in the space of binary unrooted trees with n tips $\mathcal{T}^n$. Altogether, the $n$ points are sampled in the tangent space, projected onto

the hyperboloid and combined to form a tree:

$$\left(\mathbb{T}_0\mathbb{H}^d\right)^n \xrightarrow{\phi} \left(\mathbb{H}^d\right)^n \xrightarrow{d_\kappa} \mathbb{R}^{\binom{n}{2}} \xrightarrow{\mathrm{NJ}} \mathcal{T}^n \tag{4}$$

To propose a new MCMC state, we use a multivariate Gaussian distribution centred at the current embedding locations in the tangent space, see Appendix A in S1 Text. Because the sampling space ($\mathbb{H}^d$) is different to the posterior (tree) space, a Jacobian adjustment is required, detailed in Appendix D in S1 Text. However, the transformation from the embedding space to tree space by neighbour joining has no known analytical Jacobian, which we discuss later.

## Experiments

An initial warm-up period of $10^4$ iterations serves to tune the covariance matrix before switching to the robust adaptive Metropolis (RAM) algorithm until reaching $10^6$ iterations, see Appendix A in S1 Text. Dodonaphy draws $10^4$ evenly spaced tree samples throughout the simulation. Unless otherwise stated, simulations are run in three dimensions with curvature $\kappa$ = −1.

We looked at eight biological datasets (DS1-DS8) compiled in [23] to evaluate the fidelity of hyperbolic space for Bayesian phylogenetics. The datasets contain between $n$ = 27 and $n$ = 64 taxa of aligned nucleotide sequences and are commonly used for phylogenetic benchmarking [24]. They are freely available from treebase [25] and assembled in a package stored on Dryad [26] with their identifier listed in Appendix F in S1 Text.

## Dryad DOI

10.5061/dryad.6hdr7sr3p.

## Results

To compare posterior distributions of phylogenetic trees we consider the distributions of splits, branch lengths, total tree lengths and the posterior tree probability. First, we present these in detail for DS1. The choice of curvature and embedding dimension is kept constant initially. However, we then we illustrate Dodonaphy's capacity over a range of datasets, curvatures and embedding dimensions.

### Embedding fidelity

In this section we test Dodonaphy's capability for MCMC without using an initial tree. The pairwise evolutionary distances (number of substitutions per site) between sequences are fed as a distance matrix straight into hydra+ to embed the taxa as hyperbolic points before running for a lengthier $10^7$ MCMC generations. We fix the curvature at $\kappa$ = −1 and embedding dimension $d$ = 3, choices that are informed by extensive analysis later in the text.

Fig 2A demonstrates that Dodonaphy can move from this starting state into a similar region of tree space as the golden runs of MrBayes. The starting probability for the joint distribution of the data and parameters for Dodonpahy is about −8266, whereas for MrBayes it initialises to −9822. The trace plot illustrates that the burn-in period appears completed and the MCMC does not get stuck in one state for too long, pointing to good mixing. We computed three recommended measures of effective tree sample size compared to the golden run: the Frechet Correlation ESS 198.4518, the median pseudo-ESS 364.2858 and the minimum pseudo-ESS 286.5110, see [27]. These values are below the newly supported thresholds of 500 or 625, but above (or almost above) the previously accepted threshold of 200.

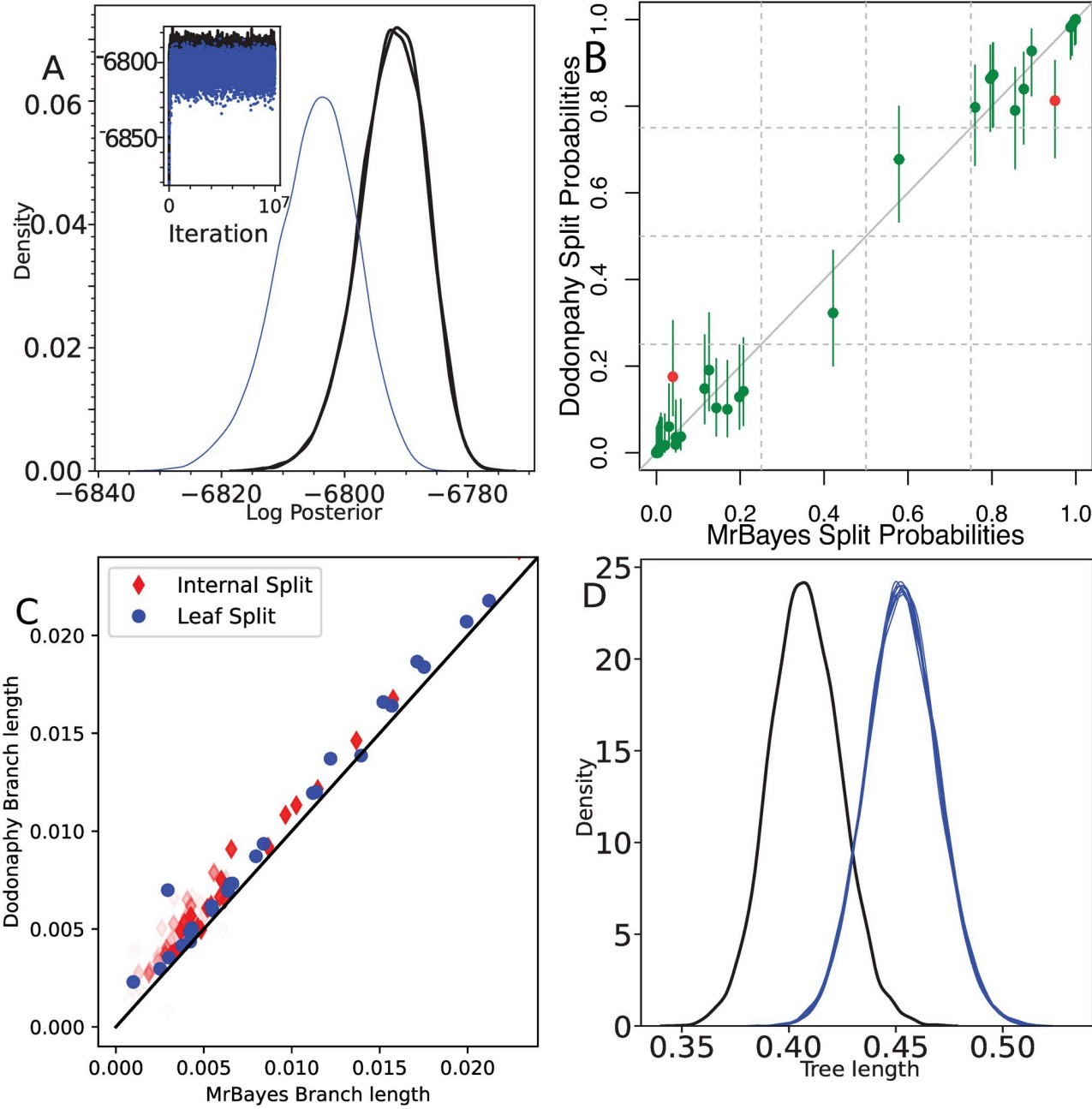

**Fig 2. Comparison between MrBayes and Dodonaphy's MCMC starting from the evolutionary distances.** Comparison of (a) posterior probability trace, (b) split frequencies with 95% confidence intervals (CI), (red dots: MrBayes value not covered by CI), (c) mean branch lengths (leaf edges in blue circles, internal edges in red diamonds), (d) total tree length estimation of 10 repeats. Markers in (c) are shaded by the frequency of appearance in the golden run. In (a, d) black lines show the two MrBayes runs and thin blue lines show Dodonaphy's results.

Dodonaphy explores the posterior splits with considerable fidelity, Fig 2B. The confidence intervals of each split's probability is computed using the Fréchet effective sample size [27]. It recovered every split appearing in the golden run with support above $10^{-3}$. Of the 64 splits from MrBayes that Dodonaphy misses, only one split was visited by either golden run more than once.

To summarise the frequency of splits appearing in the posterior, we employ the average standard deviation of split frequencies (ASDSF) [23]. It is a common statistic used to compare the splits between tree distributions, for example as used by MrBayes. A threshold of ASDSF < 0.05 is commonly used to indicate that the split frequencies from two distributions closely match each other. Indeed, the ASDSF between Dodonaphy and the first golden run is 0.003, well below the threshold for equivalence.

The mean length of splits appearing in both the golden runs and Dodonaphy are compared in Fig 2C. They generally match well, however, Dodonaphy tends to slightly overestimate branch lengths. Consequently, the mean tree length posterior is overestimated by Dodonaphy: 0.453 compared to 0.408, a ratio of 1.11. The mean variance of the total tree length matches to a high degree: a ratio of $2.830 \times 10^{-4}/2.656 \times 10^{-4} = 1.007$. Compared to the golden run, this overestimation reduces the overall log posterior, Fig 2A inset.

Repeated runs of Dodonpahy yield similar results on the tree length, Fig 2D. The support, mean location and shape of the tree length distribution are self-consistent.

## Parameter analysis

To reduce burn-in, the MCMC chains in the following sections are initialised from the consensus tree from MrBayes and run for a total of $10^6$ generations. This reduces the computational burden of running numerous MCMCs, allowing an extensive hyper-parameter analysis. To a practitioner, a more realistic starting point would be the neighbour joining tree, which is also provided as an option in Dodonaphy.

These results again report the ASDSF between Dodonaphy and the golden run. We present an alternative statistic based on the expected split frequency difference for a given effective sample size in Appendix F in S1 Text [28]. It produces similar trends to the following results with the ASDSF but shows the proportion of splits that have not yet converged to the golden run's posterior. In the limit of infinite MCMC samples, every Metropolis-Hasting sampler converges to the posterior target. The results in the following two sections demonstrate the performance of Dodonaphy for a fixed number of MCMC generations.

## Embedding curvature

The embedding curvature is a freely chosen hyper-parameter of the embedding space. Fig 3 reveals how a wide range of curvatures is suitable for all datasets. Panel a) shows that, compared to a golden run, the ASDSF falls below the threshold of 0.05 for curvatures in the range $-100 \leq \kappa \leq -1$. Similarly, the relative difference in the median total posterior tree length $\hat{l}$ to a golden run $\hat{l}_{mb}$ falls within 10% for $-100 \leq \kappa \leq -1$ range for all datasets.

As the space becomes flatter ($\kappa \rightarrow 0$), the space becomes more Euclidean. For $\kappa = 0$, we replace the hyperbolic distance with the regular Euclidean metric. These embeddings were initialised by feeding hydra+ a minuscule curvature of $10^{-10}$. Note that MCMC in Euclidean space differs from vanilla MCMC because the tips are still embedded. As the space becomes flatter, the variance of tree lengths is generally larger and the trees tend to be longer, but not monotonically. Euclidean embeddings had better tree length estimates than weakly hyperbolic embeddings. However, the ASDSF significantly worsens in this limit, indicating that the wrong trees were recovered in Euclidean space.

In the other extreme, when the curvature is below −100, the variance of tree lengths becomes smaller. MCMCs with low variance are consistent with being constrained to a local

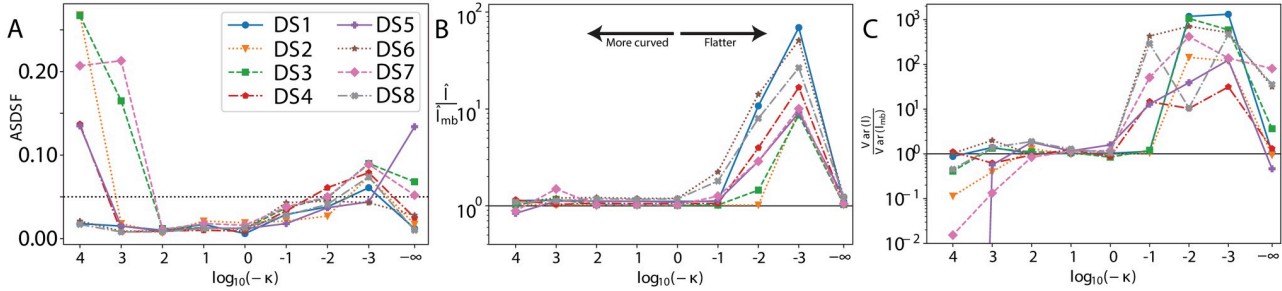

**Fig 3. Effect of embedding curvature on the posterior distribution.** Comparison to the true posterior of: a) ASDSF, b) relative difference in median tree length, c) relative difference in the variance of tree length. The truncated variance ratio for DS5 is approximately zero. The right side corresponds to flatter (and Euclidean) curvature ($\kappa = 0$ gives $\log_{10}(-\kappa) = -\infty$) and the left side is more curved.

optimum in tree space. This is also evidenced by worsening ASDSFs in this limit, signifying that, in the same data sets, the tree topologies of the full posterior are not fully explored.

## Embedding dimension

The dimension of the hyperbolic manifold is also not prescribed and may affect the quality of the MCMC. Fig 4 highlights that just three dimensions are optimal to perform Bayesian inference for all datasets analysed. Increasing beyond three dimensions does not significantly affect the splits (ASDSF). Similarly, the variance of the tree length is generally well estimated in three or more dimensions. Two dimensions appear insufficient on both fronts to capture the phylogenetic posterior.

Higher dimensions appeared to produce longer trees. This effect appears linear in the embedding dimension and depends on the data set. DS1, DS6 and DS8 are most affected,

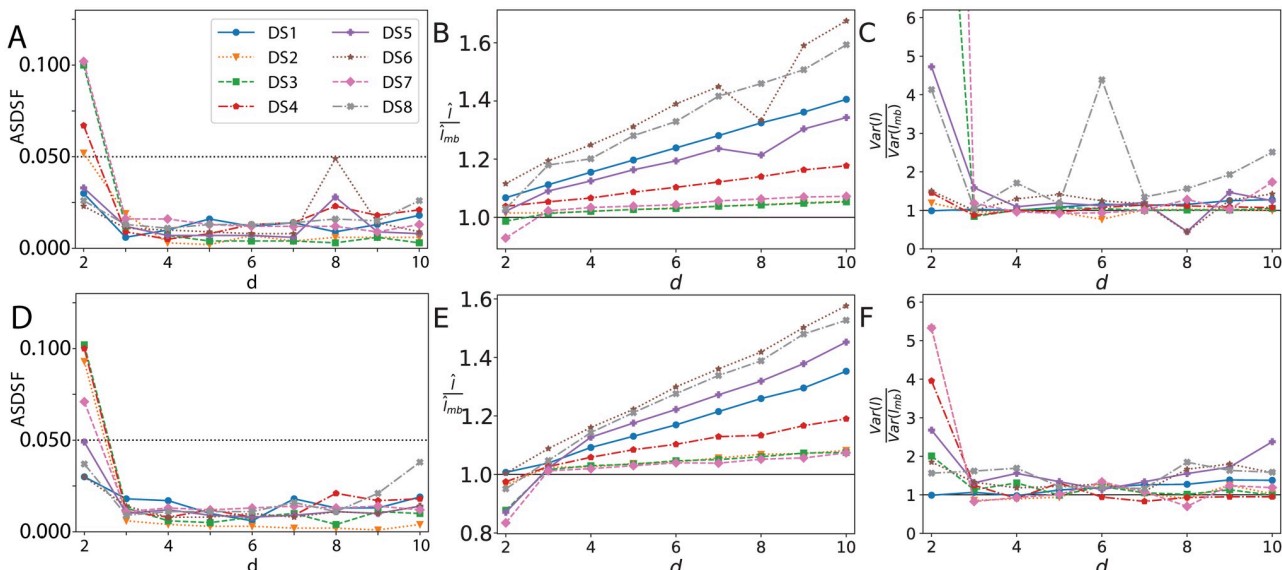

**Fig 4. Effect of embedding dimension on the posterior distribution.** Comparison to the true posterior of: a, d) ASDSF, b, e) relative difference in median tree length, c, f) relative difference in the variance of tree length. Top row curvature $\kappa = -1$, bottom row $\kappa = -100$. Truncated variance ratio in c) for DS3 with $d = 2$ is 16.41 and 39.05 for DS7.

whereas DS2, DS3, DS4 and DS7 are least affected. These groups correspond to the shortest and longest trees according to the median length from the golden runs respectively. We checked that similar results are obtained with a lower curvature (Fig 4 bottom row).

## Posterior landscape

To gain an intuition for the effect of curvature on the posterior landscape, we mapped how the posterior landscape changes by moving one taxon. We selected a set of embedding locations from an MCMC run in two dimensions $\mathbb{H}^2$. We then fixed all taxa locations except one, for which we performed a grid search. At each embedding location of this node, an NJ tree is decoded and the joint probability of this tree and the data is recorded. To plot in two dimensions, we projected the locations on the hyperboloid onto $\mathbb{R}^2$ by $\phi^{-1}$.

A heat map of the posterior in $\mathbb{H}^d$ ($d = 2$) reveals that the posterior distribution is multifaceted and depends on the locations of the other nodes, Fig 5A. It is possible to move between different topologies, however, the posterior appears multi-modal and is only smooth piecewise, with distinct regions corresponding to different tree topologies. This correspondence is apparent when compared with the middle panels, which shows the symmetric difference

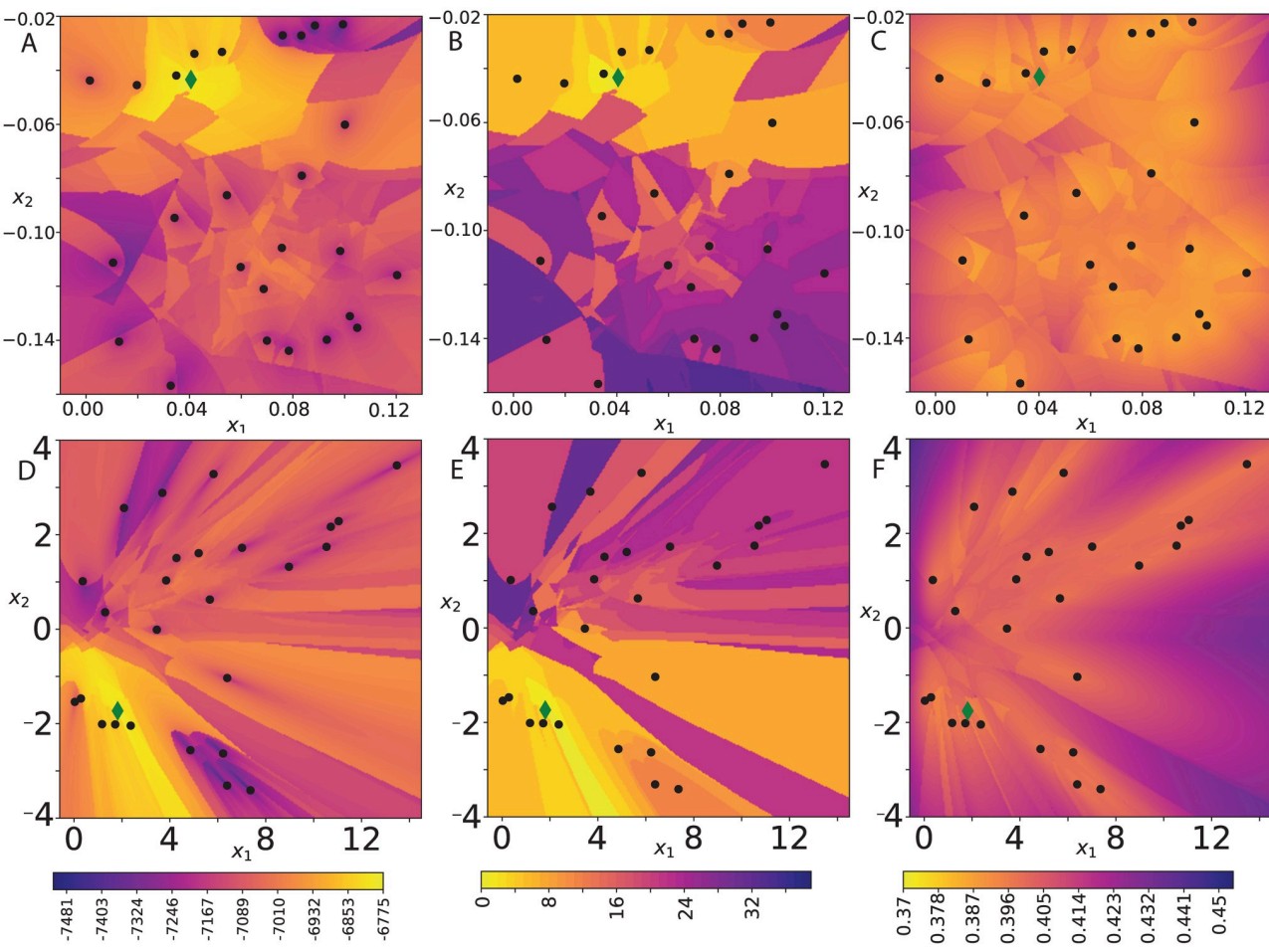

**Fig 5. Heat map of the decoded tree's properties.** Obtained by a grid search in $\mathbb{H}^2$, moving one node (green diamond) through embedding space. a, d) joint probability $p(T, Y)$, b, e) symmetric difference from best tree topology and c, f) total length. Top row curvature $\kappa = -1$ and bottom row $\kappa = -1000$. Black dots are the fixed locations of the remaining nodes.

(Robinson-Foulds distance) between the decoded tree to the optimal tree. Coarsely, locations leading to a larger symmetric difference are further in the embedding space from the optimal location.

Consistent with corollary 3, the tree length surface is smoother with lower curvature, Fig 5C and 5F. With more negative curvature $\kappa = -1000$, changes in the posterior surface are less sharp, Fig 5D. There are fewer jumps in the posterior and the regions have more uniform boundaries between them, Fig 5E. This reflects the more local nature of the very curved space, where the embedding is pushed away from the origin by the curvature. Moving one node has a smaller and more local effect (in terms of taxon-taxon distances) on the tree. Distal nodes are almost unaffected by perturbing this node. Sampling from a Normal distribution about the node could match the posterior surface better with this low curvature.

### Navigating tree space

From a given tree embedding, the topology freely changes as new locations are proposed. This alleviates the significant combinatoric barrier faced by many phylogenetic tools when changing topology. Typically, this change is done with a single NNI or an SPR move [19, 29, 30]. If the space of tree topologies is viewed as a graph with one topology on each node and NNI moves along the edges, using an embedding space allows shortcuts along the graph. On large data sets, using only NNI moves may take an unfeasible number of moves to make meaningful changes to the topology [31].

We demonstrated how using an embedding can perform multiple SPR moves at once as follows. We embedded a simulated tree with 100 taxa into $\mathbb{H}^3$ ($\kappa = -1$) and took 2000 Gaussian samples with varying covariance around this tree embedding. For each sample, we decoded an NJ tree from the locations to see how many (minimum estimated) SPR moves it was from the originally embedded tree [32], Fig 6. Using an embedding space provides a convenient way to make modifications—both continuous and discrete— to many parts of the topology at once.

### Discussion

### Fidelity of embedded MCMC

The results indicate that hyperbolic space offers fruitful embeddings for Bayesian phylogenetics. A standard MCMC algorithm can use hyperbolic space to navigate the posterior phylogenetic landscape and produce the posterior tree distribution. It can start from an embedding initialised by the NJ tree on the sequence alignment to arrive at the posterior. Once converged, it recovers similar splits frequencies, split lengths and tree lengths to the *ground-truth*.

As expected, hyperbolic embeddings performed significantly better than Euclidean embeddings. In Euclidean space, phylogenies can be embedded by taking the square root of their distances [33]. However, embedding $n$ taxa could require as many as $n - 1$ dimensions [34]. This makes the memory requirements of the embedding less scalable, going from $\mathcal{O}(ndL)$ (hyperbolic) to $\mathcal{O}(n^2L)$ (Euclidean). Consequently, Euclidean space is less suitable for embedding the large datasets being produced by infectious disease surveillance programs.

Previous work on maximum likelihood phylogenetics via hyperbolic embeddings has shown that the fidelity of the embedding improves as the embedding curvature decreases [10]. In line with this existing work but in a Bayesian context, we find that decreasing the curvature from zero improves the MCMC quality. This improvement saturates at about $\kappa = -1$, indicating that the error to the four-point condition $\delta$ is practically negligible compared to the branch lengths. Decreasing beyond this to a wide range of curvature $[-100, -1]$ provided good results, indicating the suitability of hyperbolic space for phylogenetic embeddings.

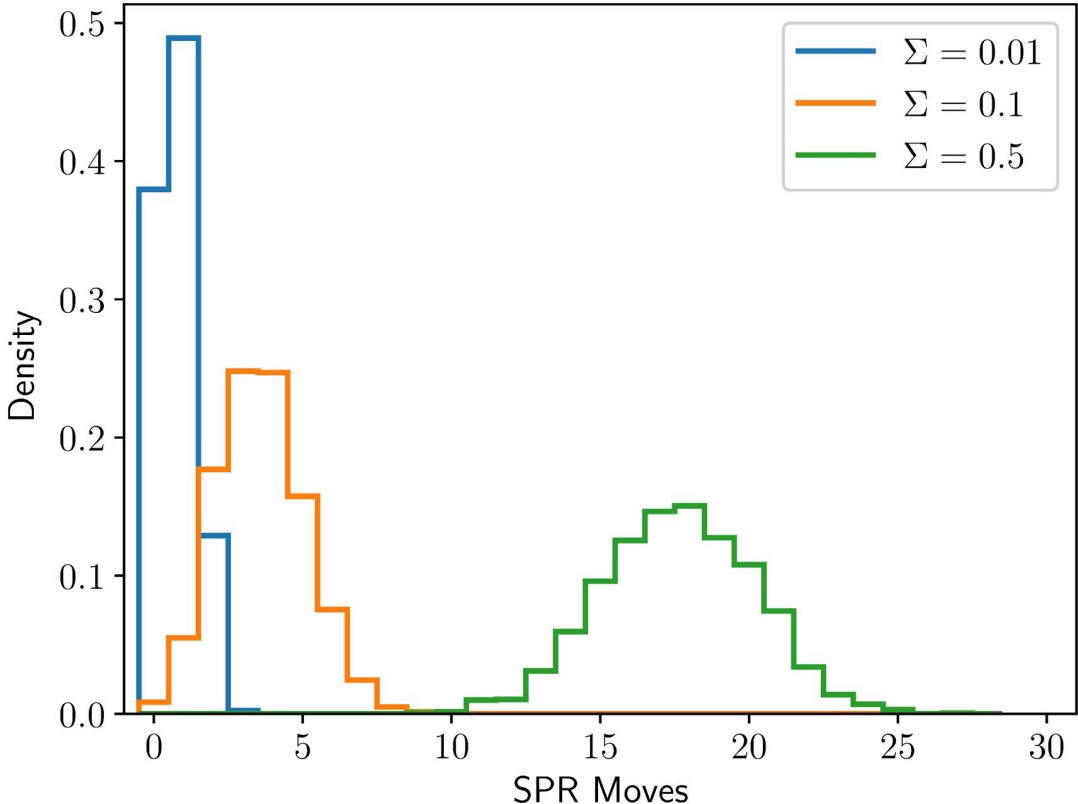

**Fig 6. SPR distances of samples about an embedded tree.** SPR distance of trees sampled from a Multivariate Gaussian (covariance $\Sigma \times I$) about a tree containing 100 leaves embedded in $\mathbb{H}^3$ with curvature $\kappa = -1$. Distributions are estimated from 2000 samples.

Lower curvatures tended to produce smaller tree length variances than the golden run. Underestimated variances could signify that the MCMC chains are contained in local optima, which is consistent with the higher split deviation (ASDSF) from the golden run. They could also arise from a systematic shift by the Jacobian not included, however, this is not consistent with the data; some datasets have similar tree length variances at low curvature Fig 3c. This suggests that there is an optimal range of curvature for embedding phylogenies, balancing between competing factors. Decreasing the curvature makes the tree length and, to some degree, the posterior smoother. However, a very curved embedding space becomes too localised and it is difficult to make significant changes to the tree topology.

Overestimated tree length variances indicate the MCMC is proposing trees outside the credible set, and sampled trees are either too long or too short compared to the credible set. Generally, this occurred for embeddings with an almost Euclidean curvature (close to zero), where the error on the four-point condition is larger. When this four-point condition error is large, neighbour joining no longer produces consistent trees, which could produce these convergence issues. With a more curved space ($\kappa = -100$) this did not occur as significantly for three or more dimensions.

Just three hyperbolic dimensions appear sufficient to continuously represent phylogenetic trees. Higher dimensions led to overestimated tree lengths, especially for short trees. This overestimation occurred regardless of the embedding curvature and appeared linear in the embedding dimension. This result highlights the suitability of low-dimensional hyperbolic

embeddings, especially given the lower run-time of low dimensions, see Appendix E in S1 Text.

One possible explanation of the tendency to overestimate branch lengths is the Jacobian involved in transforming proposal samples in hyperbolic space to samples in tree space, via neighbour joining. The Jacobian comes from each of a series of three transformations, starting from the sample space $\mathbb{R}^d$ and ending in tree space. Points are sampled in $\mathbb{R}^{nd}$ and 1) projected onto the hyperboloid $\mathbb{H}^{nd} \subset \mathbb{R}^{n(d+1)}$ via $\phi$, 2) then these points are transformed into distances in $\mathbb{R}_{\geq 0}^{\binom{n}{2}}$ before 3) finally being transformed into a tree by neighbour joining. Closed forms for the Jacobians of the first two are given in Appendix D S1 Text.

However, the determinant of the Jacobian of the second transformation is necessarily zero because for each of the $\binom{n}{2}$ the distances, only $2(d+1)$ of the inputs (the two embedding points) contribute to the distance. The derivative of each distance is zero with respect to the other $(n-2)(d+1)$ embedding locations. As we discuss below, multiple embedding configurations may produce the same set of distances, yielding a non-invertible transformation, consistent with a null Jacobian determinant. Further, computing the Jacobian of neighbour joining is non-trivial because it requires the non-differentiable min function. These Jacobian adjustments are omitted from Dodonaphy's algorithm.

One remedy for this issue is to use a prior on the embedding locations rather than on trees. We demonstrate that using a standard multivariate Normal as a prior does alleviate this trend in Appendix C in S1 Text. A drawback of the multivariate Normal prior is that it is an uncommon prior for phylogenetics, making it difficult to compare to state-of-art methods.

## Tree considerations

Embedding the internal nodes of a tree could lead to more degrees of freedom to explore tree space faster. This is simple for a fixed, given topology. However, connecting the internal nodes into a consistent bifurcating tree is a challenging problem.

Multi-furcating trees could be decoded using a (minimum) spanning tree or the approach used by SLANTIS [35], which connects internal nodes using Bernoulli trials. The efforts of [36] yielded a way to convert a multi-furcating tree into a bifurcating tree with the same likelihood. One could make use of this algorithm after forming a minimum spanning tree of the embedded points. However, the method does not maintain consistency: distances in hyperbolic space between internal nodes wouldn't translate to distances on the tree.

Dodonaphy constructs unrooted NJ trees, which is useful for exploring the relationship between taxa. If a rooted time tree is desired, other clustering algorithms such as the UPGMA or WPGMA would produce rooted trees from the taxa embeddings. However, to output a unique tree, these algorithms require ultra-metric distances, an even stronger condition than the four point condition that hyperbolic space doesn't guarantee. Ultrametric distances would permit the use of molecular clock models.

## Future research

**Beyond JC69.** The JC69 model has no free parameters, so on a given topology, the branch lengths alone determine the model's likelihood. Substitution models with free parameters, such as the generalised time-reversible substitution model [37], are not directly encoded by the embedding. Incorporating these additional model parameters could be done as in regular MCMC, rather than in an embedding space.

**Embedding isometries.** Isometries (distance preserving maps) of hyperbolic space can move an embedding to numerous locations and produce the same tree. These are a byproduct of using hyperbolic embeddings, however, they may be a useful feature for exploring the posterior landscape faster. The isometries of a tree form an equivalence class $[T]$, including the symmetries of hyperbolic space (such as rotations) as well as symmetries in the tree structure.

For example, swapping the position of two taxa in a cherry could leave the decoded tree unaltered. Knowing the isometries presents an opportunistic question: could isometric configurations of the same tree produce different neighbourhoods in tree space? This could lead to improved MCMC mixing.

**Local tree moves.** When the curvature is sufficiently small, neighbour joining decodes trees where the taxon-taxon distances in the decoded tree are arbitrarily close to the embedding distances, corollary 3. As a result, small perturbations of the locations of embedded sequences produce small changes to the taxon-taxon distances in the decoded tree. This provides a natural way to move "locally" in treespace: comparing the taxon-taxon distances, rather than the discrete number of NNI moves. Hyperbolic embeddings can also incorporate additional information about the taxa, such as their environment, morphology and gene expression, allowing for a more complete understanding of the evolutionary relationships between taxa.

Dodonaphy proposes new MCMC states by continuous probability distributions. Here it uses a projected multivariate Normal distribution for proposals, however, other distributions could equally be used. We discuss an alternative mapping that "wraps" distributions onto the hyperboloid with similar results in Appendix B in S1 Text. Matching the proposal distribution to the posterior is sought for the efficiency of MCMC chains. Exploring additional proposal distributions could lead to a better match.

This representation of the tree space opens up the possibility of exploring the neighbourhood of trees. Dodonaphy's representation of tree space bypasses classical topology changes such as NNI and SPR. Relating new moves found by Dodonaphy to existing moves (NNI, SPR) could improve methods that need to propose trees, such as MCMC.

**Adding taxa to an existing phylogeny.** Online viral phylogenetics takes genetic sequences as they arrive, i.e. sequentially. Including these new sequences into trees in the posterior space is a matter of active research [38, 39]. Dodonaphy could offer the ability to place a new taxon into a posterior distribution of trees in constant time via the embedding space. Below we outline how this can be achieved by embedding the new taxon as a distribution centred at a new embedding location around the existing embedded taxa.

One simple way to approximate the embedding location $z'$ of a new taxon labelled $k$ is to match its genetic (hamming) distance from the sequence alignment $d^*$ and hyperbolic distance $d_\kappa$ to a subset of the taxa. Select a subset $Z \subseteq X$ of taxa with cardinality one more than the embedding dimension $|Z| = d + 1$. This results in $d + 1$ equations $d_\kappa(z_i, z') = d_i^*$. Using Eq gives $z_i^\top H z' = -\cosh(\sqrt{-\kappa} d_i^*)$. To simplify, let the matrix $S$ contain the taxa locations on the hyperboloid $z_i \in \mathbb{H}^d \subset \mathbb{R}^{d+1}$ in each row $S_{i*} = z_i$. Rewritten, these equations are

$$SHz' = -\cosh(\sqrt{-\kappa} d^*),$$

where the hyperbolic cosine cosh is applied element-wise to the vector $\sqrt{-\kappa} d^*$. This equation has solution $z' = HS^{-1}\cosh(\sqrt{-\kappa} d^*)$ for the position of the new point $z' \in \mathbb{R}^{d+1}$.

This solution could be mapped onto the hyperboloid sheet by selecting the closest point, or simply adjusting the first coordinate using Eq 1. Incorporating new taxa in this way could lead to a fast method of adding taxa to a a single phylogeny, or more usefully, a distribution of phylogenies.

For more accuracy, $S$ could be extended with additional taxa to become an over-determined system and solved approximately by its least squares solution $z' = -((SH)^\mathsf{T} SH)^{-1}(SH)^\mathsf{T}\cosh(\sqrt{\kappa}D)$. Alternatively, the subset of taxa $Z$ could be reselected multiple times, effectively working as mini-batches. Then the most likely placement could be used based on the reconstructed supertree's probability.

**Differentiable embeddings.** An exciting avenue for future research is the potential for gradient-based phylogenetic methods in the embedding space. The difficulty lies in passing gradients through the non-differentiable neighbour joining algorithm. However, recent developments in soft relaxations of sorting algorithms could enable a differentiable tree decoder [40]. This would allow gradient-based Bayesian inference such as Hamiltonian Monte Carlo (HMC) and variational inference without resorting to categorical tree distributions [41, 42]. [43] propose a high dimensional $d = (2n − 3)!!$ solution for HMC, using a surrogate function to smoothly bridge the potential energy between topologies. In contrast, extending Dodonaphy with a soft version of neighbour joining offers a low-dimensional embedding space for gradient-based inference.

## Conclusion

Hyperbolic space offers a fruitful embedding space for Bayesian phylogenetics. Performing MCMC on eight data sets captured the splits (ASDSF $< 0.05$) and median tree lengths within 20% on all datasets. Embedding in three dimensions with a curvature between $-100 \le \kappa \le -1$ is sufficient to attain these results, setting a useful precedent to guide future investigations of hyperbolic embeddings in phylogenetics. The MCMC can start from an embedding of the sequence data and move throughout hyperbolic space to explore the posterior space of trees.

Phylogenetic embeddings allow tree-searching algorithms to propose new states (both topology and branch lengths) from continuous probability distributions. Embeddings offer a unique way to make "local" changes in the hyperbolic representation of tree space that could integrate into existing software.

## Supporting information

**S1 Text. Supporting text with appendices.** Appendix A: MCMC details. Appendix B: Wrapping proposal vectors onto the hyperboloid. Appendix C: Impact of Normal prior on posterior. Appendix D: Closed form pf projection Jacobians. Appendix E: Algorithm Run Time. Appendix F: Datasets. Appendix G: Effective sample size.
(PDF)

## Acknowledgments

We thank Rob Lanfear and Minh Bui for their useful comments on the paper.

Computational facilities were provided by the UTS eResearch High Performance Computer Cluster.

## Author Contributions

**Conceptualization:** Matthew Macaulay, Aaron Darling, Mathieu Fourment.

**Funding acquisition:** Mathieu Fourment.

**Investigation:** Matthew Macaulay, Mathieu Fourment.

**Methodology:** Matthew Macaulay, Aaron Darling, Mathieu Fourment.

**Software:** Matthew Macaulay.

**Supervision:** Aaron Darling, Mathieu Fourment.

**Writing – original draft:** Matthew Macaulay.

**Writing – review & editing:** Matthew Macaulay, Aaron Darling, Mathieu Fourment.

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
