## [Decision Letter · Decision Letter 0]

26 Oct 2022

Dear Dr Macaulay,

Thank you very much for submitting your manuscript "Fidelity of hyperbolic space for Bayesian phylogenetic inference" for consideration at PLOS Computational Biology.

As with all papers reviewed by the journal, your manuscript was reviewed by members of the editorial board and by several independent reviewers. In light of the reviews (below this email), we would like to invite the resubmission of a significantly-revised version that takes into account the reviewers' comments.

After carefully considering the Reviewers' comments, I would like the authors to consider resubmitting a substantially revised manuscript. All three reviewers saw value in this study, though each had their concerns. I share their concerns, with the exception of the assertion that this type of manuscript is not appropriate for PLOS Computational Biology (I believe it is appropriate). There are several major issues that must be addressed if a revised manuscript is to be acceptable for publication. First, the paper needs to be revised for general readability by a wider audience. Second, there are concerns about whether the MCMC algorithm is performing as expected. Third, the generalizability of these findings (i.e., with more biologically challenging datasets) is lacking.

We cannot make any decision about publication until we have seen the revised manuscript and your response to the reviewers' comments. A revised manuscript would certainly need to be reevaluated by external reviewers.

Sincerely,

Joel O. Wertheim

Academic Editor

PLOS Computational Biology

Lucy Houghton

Staff

PLOS Computational Biology

After carefully considering the Reviewers' comments, I would like the authors to consider resubmitting a substantially revised manuscript. All 3 three reviewers saw value in this study, though each had their concerns. I share their concerns, with the exception of the assertion that this type of manuscript for PLOS Computational Biology (I believe it is appropriate). There are several major issues that must be address if a revised manuscript is to be acceptable for publication. First, there needs to be general readability for a wider audience. Second, there are concerns about whether the MCMC algorithm is performing as expected. Third, the generalizability of these findings (i.e., with more biologically challenging datasets) is lacking. A revised manuscript would certainly need to be reevaluated by external reviewers.

Reviewer's Responses to Questions

**Comments to the Authors:**

Reviewer #1: Summary: This paper examines the use of hyperbolic embeddings for Bayesian phylogenetic inference and develops a software package which implements this technique.

Strengths: A number of common test datasets are used to study what sorts of hyperbolic spaces are best for inference, and attention is paid to how MCMC moves through the space and how the existence of the tree leaves residual discontinuities. All of this is useful for future theoretical and empirical studies.

Weaknesses: I find the presentation of material in the paper a bit disjointed and the promises of gradients (differentiable embeddings) oversold.

My overall impression is that the authors have implemented an interesting piece of software and that the paper is a useful examination of the suitability of hyperbolic spaces for Bayesian inference of phylogenies. I have attempted to suggest some reorganization that I believe would help smooth the manuscript out.

Major issues

I do not have an major issues with the paper.

Minor issues

- Some additional clarity regarding the structure of the transformations of the inferred space would be useful.

-- I find myself somewhat unclear what the point of the first paragraph in the section "Embedding Fidelity" is, and what is being said about the order of operations.

--- It is said that the authors "test Dodonaphy’s capacity for MCMC starting directly from sequence data." I am not sure what is meant. If the point is that no middle-man program is needed, and that Dodonaphy can handle the neighbor joining, likelihood, and transformations, I think that could be stated more clearly.

--- Is the order of operations to construct the initial embedding sequences->D->NJ tree->tree distance matrix->embedding or sequences->D->embedding? If the latter, does starting with a potentially non-additive embedding increase burnin?

-- I don't follow the point about local changes with curvature, though it seems important. Does not changing one node's location change all distances involving that node and thus change potentially many branch lengths, regardless of the space?

-- I find the section "MCMC Proposals" rather hard to follow. It is a mix of information about the specific proposal distribution (and its tuning) and the space of the operations. And the variable X is overloaded.

--- I would suggest focusing on the space in which moves are made (rather than specifics of the move), that is, noting that moves take place in R^nxd, for tips whose locations live in H^d, while the priors and likelihood exist in the space of trees with branch lengths.

--- I believe it would also be helpful to either briefly discuss the Jacobian here or forward-reference the fact that the Jacobian will be discussed in more detail later. The Jacobian, and the related difficulties with placing priors on branch lengths when using hyperbolic embeddings, seem to me to be an important finding of the paper that is otherwise relegated to the discussion and the supplement.

-- It is said that, "This improvement saturates at about kappa = −1, indicating that the error to the four-point condition delta is negligible compared to the branch lengths." This seems important but I am not sure what "negligible compared to the branch lengths" means.

- Choice and dimensionality of the hyperbolic space

-- The sections "Embedding Curvature" and "Embedding Dimension" are very useful and sort of undersold beforehand.

--- These sections use 8 empirical datasets and a wide range of both kappa and d (leading to something near 20 analyses per dataset I believe) to come up with good choices of the hyperbolic space. This is a very useful contribution that this paper makes to any future investigations of hyperbolic embeddings in Bayesian phylogenetics.

--- However, until the above 2 sections are read, the choices of dimension and curvature made seem rather arbitrary. I think it would be useful to state clearly in "Embedding Fidelity" that the choices of kappa and d have been informed by extensive analysis.

--- It might also bear repeating in "The Hyperboloid Model" that the choice of both kappa and d will be investigated empirically on a number of real datasets.

-- I think that the section "Tree Length Continuity" would be much more useful as a section in the methods, specifically as a subsection of "Tree Decoding." It provides links between kappa and delta that I found myself wanting when reading "Tree Decoding" and it currently interrupts the flow of the otherwise empirically-focused results.

- Superfluous text: there are several sections where the paper could be streamlined by moving material to the supplement. I do not wish to force the authors to write the paper as I would have written it, so the authors should feel free to ignore these suggestions if they do not believe them to be helpful.

-- Most of the section "Differentiable Embeddings" strikes me as extraneous because the conclusion of the first paragraph is that we are still stuck with a tree that gets in the way of derivatives. Without sketching some possible way around this, I must confess I don't see much point to the section.

-- I don't think lines 381-383 ("If the distances... substitutions per site.") are particularly necessary. If we can't guarantee ultrametric distances, then I do not see the point of clock models. And if we can, then clock models and clock rates are standard enough that a statement like "ultrametric distances would permit the use of molecular clock models" would suffice, no?

-- While I applaud the authors for being very thorough in explaining their MCMC setup, I am not sure that all of the provided information is required in the main text.

--- Beyond stating the prior, posterior, and likelihood, I think lines 99-103 can largely be removed. The Metropolis-Hastings algorithm is rather standard.

--- The section "Improved MCMC" could be moved to the supplement and replaced with a sentence like, "We use Metropolis-Coupled MCMC with 4 chains and an adaptive multivariate normal proposal." Adaptive proposals are common in all areas of MCMC and MC^3 should be familiar to most of the target audience.

- Related to the point I raised above, I find the promise in the abstract of discussing differentiable embeddings to be somewhat misleading. The discussion amounts, unless I am missing something, acknowledging that it is currently impossible. While I agree that it would be nice if we could have/use differentiable embeddings, unless the authors can sketch some solution around the problem (the tree) I would prefer the authors either remove this statement from the abstract or qualify it more transparently. "Prospects" does not convey that this is a very theoretical discussion indeed.

- In the section "Adding Taxa To an Existing Phylogeny," I am unclear as to whether this is being suggested as an algorithm to place a new sequence into an entire posterior distribution of trees (very cool!) or for point estimates (which is still useful, if perhaps at the far edge of the scope of the paper).

- Figures: there are a number of small improvements that could be made to the figures that would, in aggregate, make the paper easier to read.

-- The fonts for the figure axes are all very small and hard to read, please increase the font size by a large amount.

-- Fig 2: I think a legend in the figure explaining diamonds versus circles would help. Regardless, though, what is the difference between transparent and solid diamonds? Are the solid ones multiple runs with the exact same value?

-- Fig 2B is pretty but not as easily interpretable as a standard split-split plot, which I would advocate for (but am not requiring as a prerequisite for acceptance)

-- Fig 4: the green diamond is very hard to tell apart from the black dots, at least on my screen

-- Fig 7: the caption should include "L"

-- Fig 9 has no legend

-- For figures comparing "true" to estimated branch lengths, I think that putting truth on the x-axis would more clearly communicate that branch-lengths are being over-estimated

Typos and other small comments

Author list: Fourment is not capitalized

l 24 typo: missing period in "move Recent"

l 31 this may be overly pedantic, but there are plenty of Bayesian machine learning techniques, perhaps something like "as opposed to previous use of machine learning techniques in Bayesian phylogenetics" would be more appropriate?

l ~51 I would advocate for replacing "the posterior probability is the probability of a tree and evolutionary model parameters" with something like "the posterior distribution is the joint posterior distribution of the tree and evolutionary model parameters"

l ~51 I find the use of psi for data very jarring, especially because (upper case) Psi is often used for the tree parameter. I hate to dictate notation but I would really appreciate the use of something more standard like y (lower or upper case, bold or not) or D.

l 64 "it only computes" -> "it only requires computing"

l 92 I can see how this guarantee is helpful once we have a tree, but wouldn't we risk running into trouble if we naively used the distance matrix instead of something based on a valid tree?

l ~99 "computing the posterior" -> "sampling from/approximating the posterior"

l 103 "compose samples from the targeted posterior distribution" -> "compose autocorrelated samples from the targeted posterior distribution"

l 105 "original MCMC algorithm" -> "the original/basic/standard MH-MCMC algorithm"

l 132 broken reference

l 145 compression is such a standard approach that I don't think you have to say it is being done

l 156 I wonder if the authors might comment on their choice of the ASDSF < 0.05 threshold when they (e.g. Fourment et al. (2020)) have previously advocated for more stringent ones (in that case, RMSD < 0.01, comparable to, if more stringent than, ASDSF < 0.01)?

l 163 Given that convergence in likelihoods does not directly address convergence in treespace (e.g. Harris et al. (2021)), and that Figure 2B does imply convergence in trees, why cite Figure 2A and not 2B here?

l 272 Some readers might be more familiar with this under the name RF distance

l 291 I believe that 5.086±0.766×1e−05 ms means (5.086±0.766)×1e−05 ms? This is not immediately clear to me but it lines up with the numbers in Fig 7

l 313 I don't think I see why that in particular would be the bottleneck, would it not simply make O(dn^2) into O(n^3) like Neighbor Joining?

l 336 this sounds more like an MCMC convergence issue to me, no? shouldn't the MH step correct for that?

l 344 It seems to me that the finding that changing the proposal also ameliorates this issue bears mention here too

l 344 It seems like it should be possible to place sane priors on the pairwise distances which should also obviate the problem and might be more interpretable than Normals in the embedding space? For example, two tips can't be separated by more than n-1 edges which would induce a Gamma(branch_length_rate,scale=n-1) distribution, and must be separated by at least 2 edges, which would yield a Gamma(branch_length_rate,scale=2) distribution, so something sane would lie between those and could still be parameterized implicitly in terms of a tree length (as a Gamma tree length and Dirichlet(1) edge proportions recovers exponential distributions on the branch lengths).

l 351 extra space before "."

l 394 I don't see why this is a problem? Most phylogenetic algorithms don't distinguish left from right, including, as I understand it, NJ. But identical sequence-induced symmetries do seem relevant, and I believe at least some of these DS do include identical sequences.

l 412 It also opens up potentially easier ways to diagnose MCMC convergence, give or take the potential issues induced by the symmetries.

Fourment, Mathieu, et al. "19 dubious ways to compute the marginal likelihood of a phylogenetic tree topology." Systematic biology 69.2 (2020): 209-220.

Harrington, Sean M., Van Wishingrad, and Robert C. Thomson. "Properties of Markov chain Monte Carlo performance across many empirical alignments." Molecular biology and evolution 38.4 (2021): 1627-1640.

Reviewer #2: In this manuscript, the authors proposed a novel phylogenetic tree inference method, Dodonaphy. The first key idea of Dodonaphy is to explore phylogenetic trees in an embedded space. This approach makes it possible to perform neighborhood search, which has been done by discrete operations such as NNI, by continuous operations. The second idea is the utilization of hyperbolic space. This space is more suitable for representing phylogenetic trees than the Euclidean space. The author confirmed that Dodonaphy reconstructs phylogenetic trees that is nearly equivalent to those of MrBayes in eight different data sets.

I found that the idea is exciting, and the assertion is well confirmed in various experiments. However, the authors have neither made any new biological discoveries with this software nor presented the possibility of such discoveries. This is not consistent with the scope of the PLOS Computational Biology journal. At the very least, the authors should discuss what advantages Dodonaphy has over MrBayes, or could have in the future.

Reviewer #3: Dear authors,

I reviewed your paper with the title "Fidelity of hyperbolic space for Bayesian phylogenetic

inference". Overall, I found the paper a bit hard to digest. My background is in computational phylogenetics, so the hyperboloid model approach was very dense and challenging to digest. Similarly, I can see readers less familiar with phylogenetics to stumble completely over the very briefly mentioned but not introduced concepts (e.g., NJ, JC, NNI and SPR). For a revised version, it would definitely be very helpful if the authors could explain all necessary concept more easily.

The ADSF is not a good measurement to compare between runs. The problem is when there are many splits that are considered, some strong outliers splits are overlooked because the average of the differences is taken. However, a split could be completely present in one analysis and completely absent in another. More recent work uses either the maximum deviation from the expectation (https://doi.org/10.1111/2041-210X.13727) or the Monte Carlo error of trees (https://doi.org/10.48550/arXiv.2109.07629). It would be better to use one of these approaches instead.

What is the reason that Dodonaphy overestimates the tree length? Aren't the models identical? If not, it would be important to show an example with identical models so that you can validate your implementation against MrBayes. If the models are identical, then you must find the reason for this discrepancy because it is likely either a bug in the likelihood computation or the MCMC algorithm.

I wonder how much of these results depend on the chosen datasets. In my experience, these datasets are in fact not that challenging. MrBayes performs extremely well on these dataset, so improvements are perhaps not that necessary. This could also explain your observation of only needing few dimensions for the embedding. However, there are many challenging datasets where MrBayes fails, see for example https://doi.org/10.1093/molbev/msaa295 Perhaps you can identify a few (~5) datasets were MrBayes has problems for convergence and apply your approach instead.

Why would higher dimensions affect the tree length? Shouldn't the MCMC acceptance ratio alone influence the tree length? This sounds to me suspiciously as if there is a bug, perhaps in the likelihood if it doesn't correctly use the branch lengths.

I disagree with your discussion paragraph starting at line 323. Even if a run with a lower dimension is stuck in a local optimum, this should not happen for repeated runs (at least not the same optimum). You should convince me and other readers that the inferred tree length posterior distributions are indeed correct. Only then makes it sense to speculate in the discussion about the reason why it is overestimated or underestimated. If the embeddings are only used as MCMC proposals, then all MCMC runs should converge to exactly the same posterior distribution (if run long enough)? Can you show if they do or don't?

Unfortunately, I didn't have sufficient time to check the Jacobians that you mention, but those are good places to check if the MCMC algorithm works correctly (which I'm suspecting is not).

My final and main remark targets the actual improvement using your approach. In the introduction, you state that the current MCMC algorithms are performing poor because of the simplistic NNI and SPR algorithms. So how much does your approach improve over these well established and existing methods? It would be nice to see improvements, but I couldn't find any. Does your approach convergence faster (in terms of CPU time)?

**Have the authors made all data and (if applicable) computational code underlying the findings in their manuscript fully available?**

Reviewer #1: Yes

Reviewer #2: Yes

Reviewer #3: Yes

PLOS authors have the option to publish the peer review history of their article (what does this mean?). If published, this will include your full peer review and any attached files.

Reviewer #1: No

Reviewer #2: No

Reviewer #3: No
---

## [Decision Letter · Decision Letter 1]

20 Mar 2023

Dear Dr Macaulay,

Thank you very much for submitting your manuscript "Fidelity of hyperbolic space for Bayesian phylogenetic inference" for consideration at PLOS Computational Biology. As with all papers reviewed by the journal, your manuscript was reviewed by members of the editorial board and by several independent reviewers. The reviewers appreciated the attention to an important topic. Based on the reviews, we are likely to accept this manuscript for publication, providing that you modify the manuscript according to the review recommendations.

The reviewers and I are now satisfied with this version, provided the last few comments are fully addressed.

Sincerely,

Joel O. Wertheim

Academic Editor

PLOS Computational Biology

Lucy Houghton

Staff

PLOS Computational Biology

The reviewers and I are now satisfied with this version, provided the last few comments are fully addressed.

Reviewer's Responses to Questions

**Comments to the Authors:**

Reviewer #1: I find this revised edition of the paper significantly easier to follow, and it much more clearly highlights the significant effort the authors have invested in this work. The reorganization makes it much faster to read, and I find the revised Algorithm 1 particularly helpful. I greatly appreciate the increased exploration of MCMC mixing and the exploration of the set of splits encountered.

I have a few small comments which follow, but no serious objections to the manuscript as-is.

I believe there is a typo in Algorithm 1, line 6. Should not "Pair-wise alignment distances" be "Pair-wise hyperbolic distances" as on line 10?

In the section "Dodonaphy Algorithm":

- While I think I get the broad strokes from Equation 4, I do not quite follow what exactly a tangent space of the origin is and what the new T is.

- For readers like me who are not well-versed in hyperbolic geometry, I wonder if an extra sentence of explanation of Equation 3 could be added.

- Additionally, the use of T as a tangent space, where it had been a tree, and the introduction of \\mathcal{T} for the tree is jarring.

Figure 2:

- I like the replacement for panel B and find it much more interpretable.

- The transparent diamonds, which were visible in the previous version of the MS/figure, seem to have disappeared?

- Color labels for A/D would be nice (also for the similar figure in the appendix)

- Something seems to have gone wrong with the bar coloring for the two red points in B (also in the appendix)

Circa line 440: Not every reader may be familiar with cosh(), or cosh() of a matrix.

I'm probably missing something obvious, but I'm having some trouble following the dimensions int the paragraph between 439 and 440. D seems to be (d+2)x(d+2), while S is (d+1)x(d+1). Also, does "S_{ij} = z_i" mean "the rows of S are z"?

In the text of the appendix, most references to Figure/Fig X are lower case, which might be a typo.

**Have the authors made all data and (if applicable) computational code underlying the findings in their manuscript fully available?**

Reviewer #1: Yes

PLOS authors have the option to publish the peer review history of their article (what does this mean?). If published, this will include your full peer review and any attached files.

Reviewer #1: No

Figure Files:

Data Requirements:

Reproducibility:

References:

---

## [Editor Report · Decision Letter 2]

8 Apr 2023

Dear Dr Macaulay,

We are pleased to inform you that your manuscript 'Fidelity of hyperbolic space for Bayesian phylogenetic inference' has been provisionally accepted for publication in PLOS Computational Biology.

Best regards,

Joel O. Wertheim

Academic Editor

PLOS Computational Biology

Lucy Houghton

Staff

PLOS Computational Biology

---

## [Editor Report · Acceptance letter]

20 Apr 2023

PCOMPBIOL-D-22-01264R2 

Fidelity of hyperbolic space for Bayesian phylogenetic inference

Dear Dr Macaulay,

I am pleased to inform you that your manuscript has been formally accepted for publication in PLOS Computational Biology. Your manuscript is now with our production department and you will be notified of the publication date in due course.

With kind regards,

Zsofia Freund
